# Data-Calibrated Robust Multi-Criteria Decision Support for Marketplace Logistics under Joint Uncertainty in Performance and Preferences

**Timur M. Panesh**

Anna V. Kovalenko
Kuban State University, Krasnodar, Russia

## Abstract

Selecting logistics operators and inbound channels for marketplace fulfillment is a high-impact multi-criteria decision problem under uncertainty. In practice, both criterion values and decision-maker preferences drift over time. We study additive MCDA with *joint* uncertainty in the performance matrix $M$ and weight vector $w$. We model uncertainty as a product set $\mathcal{U} = \mathcal{M} \times \mathcal{W}$, where $\mathcal{M}$ is interval (box) uncertainty around nominal KPI estimates and $\mathcal{W}$ is an $\ell_1$-ball around nominal preferences on the simplex. For finite scenario approximations, we derive exact single-level LP reductions for two robust objectives: (i) robust max–min utility and (ii) minimax regret. The resulting formulations scale linearly with the number of scenarios and provide a practical optimization core for AI-assisted decision support. Monte Carlo experiments under base and stress uncertainty regimes show stable out-of-sample gains in worst-case utility for robust max–min and systematic control of extreme losses for minimax regret. We operationalize the method in a data-driven *predict–estimate–optimize–monitor* pipeline, where uncertainty sets are learned and recalibrated from forecast residuals and preference-drift signals. This learning-to-robustness loop links model updates, robust optimization, and monitoring diagnostics in one deployable decision workflow.

## 1 Introduction

Marketplace logistics decisions jointly affect cost-to-serve, SLA compliance, delivery reliability, and disruption resilience. At marketplace scale, even modestly suboptimal operator allocation can create persistent losses and service instability. This makes operator/channel allocation a high-stakes multi-criteria decision problem, not a static ranking exercise.

A central challenge is *double uncertainty*. First, KPI estimates are noisy and delayed. Second, preference weights over criteria are often incomplete and may drift due to policy changes, seasonality, or strategic shifts. Most MCDA pipelines assume fixed $(M, w)$, while many robust variants model uncertainty in only one component. In practice, this mismatch can either overfit to nominal estimates or become overly conservative.

This paper develops a tractable robust MCDA framework for simultaneous uncertainty in criterion values and preferences. Our technical and empirical contributions are:

- We define a product-set uncertainty model $\mathcal{U} = \mathcal{M} \times \mathcal{W}$ tailored to marketplace logistics, combining box uncertainty for $M$ and simplex-constrained $\ell_1$ uncertainty for $w$.
- For finite scenario approximation, we provide exact LP reformulations for robust max–min utility and minimax regret, with linear growth in scenario count.
- We introduce an interpretable diagnostic layer (robustness bands, regret profiles, EVPI, and value of uncertainty reduction) that supports data-improvement prioritization.
- We evaluate nominal, robust max–min, and minimax regret strategies in repeated Monte Carlo train/test experiments (base and stress regimes), showing consistent trade-offs aligned with each objective.

The methodology is designed as an optimization core for AI-assisted decision support systems (DSS), where predictive modules update uncertainty descriptions and optimization modules produce robust allocations.

## 2 RELATED WORK

MCDA methods such as AHP, TOPSIS, and VIKOR remain widely used for logistics and supply-chain choices, but classical formulations are largely deterministic and sensitive to weight specification (Sahoo et al., 2023). Preference-robust portfolio-style MCDA and related robust preference programming address incomplete or imprecise weights, yet often abstract from explicit uncertainty in alternative-level KPI estimates (Liesiö et al., 2007). In parallel, robust optimization provides a principled treatment of uncertainty sets and tractable formulations for worst-case design (Ben-Tal et al., 2009; Bertsimas et al., 2011).

Recent logistics and supply-chain literature increasingly uses robust and distributionally robust optimization for network-level planning (Chen & Chen, 2025; Ash et al., 2022), but the decision structure there differs from additive MCDA with explicit preference uncertainty. In robust MCDM, recent models improve stability in conflicting stakeholder settings (D'Agostino et al., 2024; Paradowski et al., 2025), but exact tractable treatment of *joint* uncertainty in both $M$ and $w$ remains comparatively underdeveloped for marketplace operator allocation. Neural-network-based analytics for e-commerce settings has also been discussed in the applied literature (Panesh et al., 2024). This line is relevant here because neural-network-based e-commerce analytics can serve as the predictive front-end that produces forecast residuals, which are translated into interval radii $\Delta_{ij}$ and preference-drift budgets $\varepsilon$ in the uncertainty sets.

Our work addresses this gap through a product-set formulation that preserves interpretability and supports exact LP reductions under scenario approximation. Full extended related-work discussion and comparison details are moved to Appendix E due to main-text page limits.

## 3 PROBLEM SETUP

Let $I = \{1, \ldots, n\}$ be alternatives (logistics operators/inbound channels) and $J = \{1, \ldots, m\}$ criteria. A strategy is a mixture/allocation vector $x \in \mathbb{R}^n_+$ in a nonempty compact polyhedron:

$$\mathcal{X} = \left\{ x \in \mathbb{R}^n_+ \ \middle| \ Ax \le b, \ \sum_{i=1}^n x_i = 1 \right\}. \tag{1}$$

For each criterion $j$, let $u_j : \mathbb{R} \to [0, 1]$ be a monotone linear or piecewise-linear partial utility. Let $M = (m_{ij}) \in \mathbb{R}^{n \times m}$ be the KPI matrix and $w \in \mathbb{R}^m_+$ criterion weights with $\sum_j w_j = 1$. Alternative and strategy utilities are:

$$v_i(M, w) = \sum_{j=1}^m w_j u_j(m_{ij}), \tag{2}$$

$$U(x, M, w) = \sum_{i=1}^n x_i v_i(M, w) = \sum_{i=1}^n \sum_{j=1}^m x_i w_j u_j(m_{ij}). \tag{3}$$

**Modeling assumptions.** We use additive independence between criteria (standard in value-focused MCDA (Keeney & Raiffa, 1993)), piecewise-linear criterion utilities after normalization, and polyhedral feasibility constraints that capture operational policy limits.

## 4   JOINT UNCERTAINTY VIA PRODUCT SETS

Given nominal estimates $(\hat{M}, \hat{w})$, we define:

$$\mathcal{M} = \left\{M \mid \hat{m}_{ij} - \Delta_{ij} \le m_{ij} \le \hat{m}_{ij} + \Delta_{ij}, \ \forall i, j\right\}, \tag{4}$$

$$\mathcal{W} = \left\{w \in \mathbb{R}^m_+ \ \middle| \ \|w - \hat{w}\|_1 \le \varepsilon, \ \sum_{j=1}^{m} w_j = 1, \ 0 \le \varepsilon < 2\right\}, \tag{5}$$

$$\mathcal{U} = \mathcal{M} \times \mathcal{W}. \tag{6}$$

This structure separates KPI and preference uncertainty while keeping each component interpretable. Parameter $\Delta_{ij}$ controls KPI uncertainty radii, and $\varepsilon$ controls preference ambiguity. The bound $\varepsilon < 2$ avoids a trivial simplex-wide ambiguity set. In deployment, both are updated by predictive diagnostics and calibration.

The robust max–min problem is

$$\max_{x \in \mathcal{X}} \min_{(M,w) \in \mathcal{U}} U(x, M, w). \tag{7}$$

We also consider regret-based robustness. For a realized $(M, w)$, define benchmark value $V^\star(M, w) = \max_{z \in \mathcal{X}} U(z, M, w)$, and regret $R(x; M, w) = V^\star(M, w) - U(x, M, w)$. The robust minimax regret objective is $\min_{x \in \mathcal{X}} \max_{(M,w) \in \mathcal{U}} R(x; M, w)$ (Savage, 1951).

## 5   SCENARIO APPROXIMATION AND EXACT LP REFORMULATIONS

Let $\mathcal{U}_S = \{(M^{(s)}, w^{(s)})\}_{s=1}^{S}$ be a finite scenario approximation of $\mathcal{U}$. For each scenario,

$$c_i^{(s)} = \sum_{j=1}^{m} w_j^{(s)} u_j\left(m_{ij}^{(s)}\right), \qquad U(x, M^{(s)}, w^{(s)}) = (c^{(s)})^\top x. \tag{8}$$

With precomputed $u_j(m_{ij}^{(s)})$, each scenario utility is linear in $x$.

**Remark (scenario approximation and SAA).**   The finite scenario model is closely related to sample average approximation (SAA) in stochastic programming: under standard regularity conditions, as $S \to \infty$, scenario-based solution/value estimates converge to their population counterparts. In practice, we therefore complement in-sample optimization with independent out-of-sample validation (Birge & Louveaux, 2011).

### 5.1   ROBUST MAX–MIN UTILITY

The scenario counterpart of equation 7 is $\max_{x \in \mathcal{X}} \min_{s=1,\dots,S} (c^{(s)})^\top x$.

**Lemma 1** (Exact LP for scenario robust max–min). *The problem is equivalent to*

$$\max_{x \in \mathcal{X}, \eta} \ \eta \quad s.t. \quad (c^{(s)})^\top x \ge \eta, \ \ s = 1, \dots, S. \tag{9}$$

**Proof sketch.**   Introduce epigraph variable $\eta$ for the inner minimum. Constraint $\eta \le (c^{(s)})^\top x$ for all scenarios is equivalent to $\eta \le \min_s (c^{(s)})^\top x$. Maximizing $\eta$ gives equation 9. All constraints are linear in $(x, \eta)$; details are in Appendix B.

### 5.2   MINIMAX REGRET

Define per-scenario optimal values

$$V_s^\star = \max_{x \in \mathcal{X}} (c^{(s)})^\top x, \quad s = 1, \dots, S, \tag{10}$$

obtained by solving $S$ LPs. Scenario regret is $R(x; s) = V_s^\star - (c^{(s)})^\top x$.

**Lemma 2** (Exact LP for scenario minimax regret). *The scenario minimax regret problem* $\min_{x \in \mathcal{X}} \max_s R(x; s)$ *is equivalent to*

$$\min_{x \in \mathcal{X}, \rho} \rho \quad s.t. \quad V_s^\star - (c^{(s)})^\top x \le \rho, \quad s = 1, \dots, S. \tag{11}$$

**Proof sketch.** Introduce epigraph variable $\rho$ for the maximum regret. Constraints enforce $\rho \ge R(x; s)$ scenario-wise, which is equivalent to $\rho \ge \max_s R(x; s)$. Minimization yields equation 11; full derivation is in Appendix B.

**Complexity and tractability.** Both LPs have $n + 1$ decision variables plus $S$ scenario constraints (and constraints defining $\mathcal{X}$), hence linear scaling in $S$ after computing $\{c^{(s)}\}$ and $\{V_s^\star\}$. Standard LP complexity bounds apply (Boyd & Vandenberghe, 2004; Bertsimas & Tsitsiklis, 1997).

# 6 DIAGNOSTICS FOR AI-ASSISTED DSS

Robust optimization outputs a strategy, but operational use also needs interpretable sensitivity diagnostics.

**Robustness band.** For fixed $x$, scenario spread is

$$\Delta(x) = \max_s (c^{(s)})^\top x - \min_s (c^{(s)})^\top x, \tag{12}$$

which approximates utility volatility under uncertainty.

**Expected value of perfect information (EVPI).** Given scenario probabilities $\{p_s\}$,

$$\text{VPI} = \sum_{s=1}^{S} p_s V_s^\star, \quad \text{VSP}(x) = \sum_{s=1}^{S} p_s (c^{(s)})^\top x, \quad \text{EVPI}(x) = \text{VPI} - \text{VSP}(x). \tag{13}$$

Large EVPI indicates high potential value from better state information (Keeney & Raiffa, 1993).

**Value of uncertainty improvement (VoIU).** If monitoring and data investment shrink uncertainty to $\mathcal{U}' \subseteq \mathcal{U}$, define

$$\text{VoIU} = V_{\text{rob}}(\mathcal{U}') - V_{\text{rob}}(\mathcal{U}) \ge 0, \tag{14}$$

with monotonicity under set inclusion. VoIU supports prioritizing which uncertainty source to reduce first.

## 6.1 CALIBRATION AND MODEL-SELECTION PROTOCOL

In practice, uncertainty radii and preference ambiguity are hyperparameters, not fixed constants. We therefore use a calibration protocol aligned with deployment:

1. Build nominal forecasts $(\hat{M}, \hat{w})$ and uncertainty candidates $\{\Delta^{(k)}\}$, $\{\varepsilon^{(\ell)}\}$ from historical forecast errors and governance constraints.
2. For each candidate pair $(\Delta^{(k)}, \varepsilon^{(\ell)})$, solve robust max–min and minimax regret LPs on train scenarios.
3. Evaluate out-of-sample lower-tail utility and high-quantile regret on independent test scenarios.
4. Select the operating point by business objective: guarantee floor (max–min), benchmark-loss protection (minimax regret), or mixed criterion.

This procedure avoids fixing conservatism a priori and makes the choice of $\varepsilon$ auditable.

## 6.2 TWO USEFUL THEORETICAL PROPERTIES

The product-set model gives simple guarantees used in monitoring.

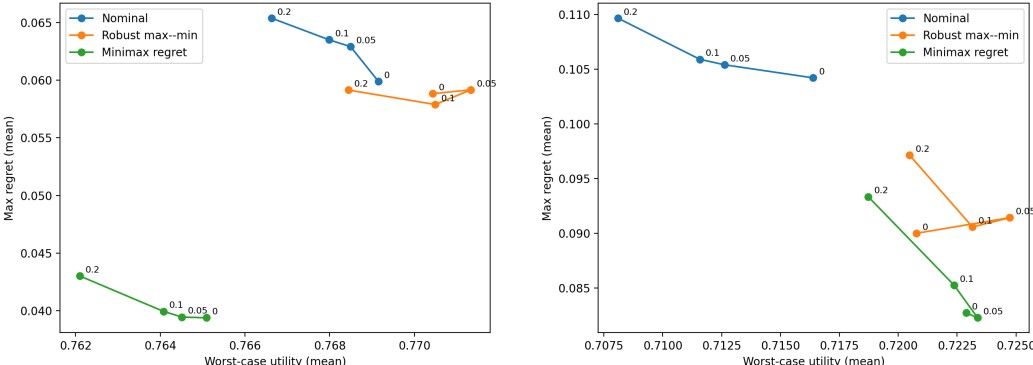

Figure 1: Out-of-sample trade-off between worst-case utility and maximum regret for base (left) and stress (right) regimes. Marker labels denote $\varepsilon \in \{0, 0.05, 0.10, 0.20\}$.

Table 1: Monte Carlo results (mean [95% bootstrap CI], $R = 100$ runs). Each cell reports **worst-case utility** and **max regret**.

| Mode | $\varepsilon$ | Nominal | Robust max–min | Minimax regret |
|---|---|---|---|---|
| *base* | | | | |
| base | 0.05 | 0.779 [0.778; 0.780] / 0.047 [0.047; 0.048] | 0.780 [0.779; 0.781] / 0.047 [0.046; 0.048] | **0.782 [0.781; 0.783] / 0.038 [0.037; 0.038]** |
| base | 0.10 | 0.776 [0.775; 0.777] / 0.049 [0.049; 0.050] | 0.777 [0.776; 0.778] / 0.049 [0.047; 0.050] | **0.780 [0.779; 0.781] / 0.039 [0.038; 0.039]** |
| base | 0.20 | 0.768 [0.766; 0.769] / 0.059 [0.058; 0.061] | 0.770 [0.768; 0.771] / 0.057 [0.055; 0.059] | **0.773 [0.772; 0.774] / 0.045 [0.045; 0.046]** |
| *stress* | | | | |
| stress | 0.05 | 0.743 [0.741; 0.744] / 0.089 [0.088; 0.090] | 0.750 [0.748; 0.751] / 0.084 [0.083; 0.086] | **0.752 [0.750; 0.753] / 0.074 [0.073; 0.076]** |
| stress | 0.10 | 0.740 [0.738; 0.742] / 0.090 [0.089; 0.091] | 0.748 [0.746; 0.749] / 0.086 [0.084; 0.087] | **0.749 [0.748; 0.751] / 0.075 [0.074; 0.077]** |
| stress | 0.20 | 0.731 [0.729; 0.733] / 0.099 [0.097; 0.100] | 0.740 [0.738; 0.742] / 0.092 [0.090; 0.094] | **0.743 [0.742; 0.745] / 0.081 [0.080; 0.083]** |

**Remark.** If $\mathcal{U}_1 \subseteq \mathcal{U}_2$, then $V_{\mathrm{rob}}(\mathcal{U}_1) \geq V_{\mathrm{rob}}(\mathcal{U}_2)$; shrinking uncertainty cannot decrease robust value.

**Remark.** For fixed $x$ and scenario set $\mathcal{U}_S$, the quantity $\max_s R(x; s) = \max_s \left[ V_s^\star - (c^{(s)})^\top x \right]$ is exactly the LP equation 11 epigraph value with fixed $x$, so the same constraints support optimization and ex-post diagnostics.

## 7 EXPERIMENTS

### 7.1 DESIGN

We compare three strategies: nominal optimization at $(\hat{M}, \hat{w})$, robust max–min, and minimax regret. Criteria include cost, lead time, reliability, defect level, volatility, and disruption risk. Criterion values are normalized to $[0, 1]$ with direction-consistent scaling.

Two uncertainty regimes are tested: *base* (moderate uncertainty) and *stress* (amplified KPI and preference variability). Preference uncertainty uses $\|w - \hat{w}\|_1 \leq \varepsilon$ for $\varepsilon \in \{0, 0.05, 0.10, 0.20\}$.

For each regime and $\varepsilon$, we run repeated train/test Monte Carlo: $R = 100$ independent repetitions, $S_{\mathrm{train}} = 500$ scenarios for optimization, and $S_{\mathrm{test}} = 5000$ independent scenarios for out-of-sample evaluation. Confidence intervals are estimated by nonparametric bootstrap over repetitions ($B = 5000$) (Rubinstein & Kroese, 2016; Efron & Tibshirani, 1993; Davison & Hinkley, 1997).

### 7.2 MAIN EMPIRICAL FINDINGS

Figure 1 summarizes the key structural result in both uncertainty regimes: stable trade-off frontiers between worst-case utility and maximum regret across strategies and $\varepsilon$.

Across regimes, robust max–min provides the strongest guarantees on the lower tail of utility, while minimax regret systematically controls benchmark-relative losses. Under stress, differences between robust and nominal strategies increase, indicating practical value of explicit uncertainty modeling.

**Interpretation by decision objective.** The empirical pattern is consistent with objective design. Max–min optimization shifts allocations toward alternatives with smaller downside exposure under adverse combinations of KPI and preference shocks; this improves floor performance at the cost of potentially higher benchmark-relative loss in favorable states. Minimax regret instead controls distance to scenario-wise clairvoyant benchmarks, leading to more balanced performance when the true state is uncertain and adversarially variable.

**Stability across uncertainty levels.** Increasing $\varepsilon$ acts as a robustness dial. In both base and stress regimes, strategy trajectories remain ordered: nominal solutions are less conservative, robust max–min moves toward higher guarantees, and minimax regret tracks lower extreme benchmark loss. This stable ordering is valuable operationally because it enables policy selection by risk appetite rather than one-off tuning.

**Why stress testing matters.** Base-regime performance alone may hide fragility. Stress scenarios amplify simultaneous perturbations in $M$ and $w$, where nominal allocations exhibit wider downside spread. Robust formulations are most informative exactly in this regime, which better reflects holiday peaks, provider disruptions, and sudden policy shifts.

**Quantitative reading of Table 1.** The numeric table shows two persistent patterns. First, for each $(\text{mode}, \varepsilon)$ pair, minimax regret is the best method on the max-regret metric, with clear gaps to nominal policies (e.g., in stress at $\varepsilon = 0.20$, regret decreases from $0.099$ to $0.081$). Second, robust max–min consistently improves worst-case utility over nominal under stress (e.g., at $\varepsilon = 0.20$, from $0.731$ to $0.740$), while keeping regret substantially below nominal.

At moderate ambiguity ($\varepsilon \in \{0.05, 0.10\}$), confidence intervals are narrow and ordering is stable across runs, indicating that the observed trade-off is not driven by isolated outliers. This is operationally relevant because policy ranking remains predictable under repeated retraining and independent scenario draws.

**Inter-regime robustness gap.** Comparing base and stress regimes at the same $\varepsilon$ highlights how uncertainty amplification changes policy value. Nominal allocations degrade more strongly in both utility floor and regret tail, whereas robust objectives preserve more stable behavior. For example, when moving from base to stress at $\varepsilon = 0.10$, nominal worst-case utility drops by about $0.036$ (from $0.776$ to $0.740$), while robust max–min drops by about $0.029$ (from $0.777$ to $0.748$). This relative gap quantifies the resilience benefit of explicit robustness.

**Runtime and scalability in practice.** For each configuration, minimax regret requires precomputing $V_s^\star$ through $S$ LP solves, then one master LP; robust max–min requires one master LP after scenario coefficient preprocessing. In our Monte Carlo setup ($S_{\text{train}} = 500$), this workload remains practical for periodic decision cycles and scales approximately linearly with $S$. In production scheduling, this means scenario budgets can be increased in stress periods with predictable runtime growth rather than abrupt computational bottlenecks.

**What is moved out of main text.** To satisfy the strict page policy, we move full numeric result matrices, additional plots (worst utility and max-regret curves by regime), and extended calibration details to appendices after references.

### 7.3   Detailed reading of the base–stress comparison

The base regime represents normal operational volatility. In this regime, the nominal strategy can remain competitive on central tendency, but its lower-tail behavior is less stable because it does not hedge against joint perturbations in both $M$ and $w$. Robust max–min strategies intentionally sacrifice some optimistic-state performance to preserve a stronger floor across scenarios. Minimax regret strategies are less floor-oriented and instead align with benchmark-relative protection.

The stress regime sharpens these differences. When KPI perturbation radii and preference ambiguity increase simultaneously, nominal allocations become more sensitive to adverse scenario combinations. The robust max–min policy maintains a more stable lower bound, while minimax regret exhibits the tightest control of maximum benchmark-relative loss. This regime is especially relevant for marketplace peaks, disruption cascades, and short-term policy shifts where historical averages are weak predictors.

## 7.4 SENSITIVITY TO PREFERENCE AMBIGUITY RADIUS $\varepsilon$

The parameter $\varepsilon$ has a direct managerial interpretation as tolerated preference ambiguity. Small $\varepsilon$ keeps solutions close to nominal preference elicitation, while larger values enforce hedging against preference drift.

Empirically, increasing $\varepsilon$ produces smooth strategy trajectories rather than discontinuous jumps. This is important in production settings because abrupt allocation changes can create operational frictions (capacity re-contracting, route updates, and renegotiation overhead). The observed smoothness supports using $\varepsilon$ as a governed policy dial:

- lower $\varepsilon$ for stable periods with strong confidence in preference elicitation,
- higher $\varepsilon$ for volatile periods or organizational re-prioritization,
- intermediate $\varepsilon$ when balancing service-level guarantees and opportunity-loss protection.

## 7.5 SCENARIO COUNT AND COMPUTATIONAL BEHAVIOR

Let $r$ denote the number of linear constraints defining $\mathcal{X}$. For fixed $n$, both LP formulations add $S$ scenario constraints and one epigraph variable; minimax regret additionally needs preprocessing $V_s^\star$ via $S$ LP solves. Therefore, end-to-end effort grows approximately linearly in $S$ after scenario generation and utility preprocessing.

This linear growth is central for DSS integration: one can allocate larger scenario budgets to stress periods without changing model class, and can parallelize preprocessing over scenarios. In operational deployments, this enables predictable runtime governance and reproducible schedule planning for recurring decision cycles.

# 8 PRACTICAL DEPLOYMENT GUIDANCE

## 8.1 FROM PREDICTIVE MODELS TO OPTIMIZATION-READY UNCERTAINTY

In marketplace systems, uncertainty descriptions are produced upstream by forecasting and analytics modules. The translation into optimization-ready sets follows three steps.

**Step 1: KPI uncertainty extraction.** For each criterion and alternative, prediction residuals (or calibrated quantile intervals) define candidate interval radii $\Delta_{ij}$. Criteria with weaker data quality naturally obtain wider intervals, yielding larger protection against adverse realizations.

**Step 2: Preference uncertainty extraction.** Preference vectors can be obtained from expert elicitation, policy priorities, or implicit behavioral signals. Disagreement or temporal drift in these signals is mapped into an $\ell_1$ ambiguity radius $\varepsilon$ around the nominal weight vector.

**Step 3: Governance constraints and feasibility.** Operational rules (capacity caps, minimum service shares, contractual bounds, diversification constraints) populate $Ax \leq b$ in $\mathcal{X}$. This keeps robust optimization aligned with implementable allocations.

## 8.2 DECISION PROTOCOL FOR STRATEGY SELECTION

A practical policy should not choose between max–min and minimax regret in the abstract; it should tie objective choice to risk posture and business penalties.

- If SLA-violation cost or failure externalities dominate, choose robust max–min as the default policy and monitor regret as a diagnostic.
- If management emphasizes closeness to scenario-wise best feasible decisions, choose minimax regret and monitor floor utility as a safety constraint.
- If both concerns are critical, maintain a dual-dashboard policy where strategy updates require acceptable values for both worst-case utility and worst-case regret.

### 8.3 DIAGNOSTICS-DRIVEN DATA INVESTMENT

EVPI and VoIU provide complementary guidance for data-improvement priorities. High EVPI indicates potential benefit from improved state information in general; high VoIU for a specific uncertainty reduction action indicates direct robust-value gain from shrinking a concrete uncertainty source.

In practice, one can rank candidate interventions (better lead-time sensing, defect-data cleaning, supplier reliability audits, preference re-elicitation) by estimated VoIU per unit cost. This turns robustness diagnostics into an actionable investment queue rather than a passive reporting layer.

## 9 LIMITATIONS AND RESEARCH DIRECTIONS

The current framework deliberately prioritizes tractability and interpretability. Several extensions are important:

1. **Coupled uncertainty.** Product sets ignore explicit dependence between KPI shocks and preference changes. Coupled sets or copula-informed scenario generation are a natural next step.
2. **Distributional robustness.** Scenario sets can be embedded into ambiguity-set DRO constructions to combine worst-case guarantees with distributional calibration.
3. **Richer preference models.** Additive utility may miss interactions or threshold effects. Nonadditive extensions can improve behavioral fidelity at increased computational cost.
4. **Adaptive sequential control.** In dynamic operations, repeated re-optimization with receding-horizon updates can connect robust MCDA to online learning and control.

These directions preserve the paper's core message: robustness is most useful when uncertainty, optimization, and monitoring are jointly designed rather than treated as separate modules.

## 10 DISCUSSION: AI-ASSISTED *Predict–Estimate–Optimize–Monitor*

The framework is intended for an iterative DSS cycle:

1. **Predict/Estimate.** Forecasting modules update nominal KPIs and uncertainty radii $\Delta_{ij}$ from demand, seasonality, and supplier signals.
2. **Optimize.** LP formulations equation 9 and equation 11 produce robust allocations under explicit uncertainty assumptions.
3. **Monitor/Diagnose.** Robustness bands, regret profiles, EVPI, and VoIU identify whether performance risk comes mainly from KPI noise or preference ambiguity.
4. **Refine data policy.** High VoIU dimensions become priorities for data-collection or model-improvement investments.

Concretely, an ML predictor maps features $z$ to KPI point forecasts $\hat{m}_{ij}(z)$ and residuals $r_{ij} = m_{ij} - \hat{m}_{ij}(z)$; we set interval radii by calibrated residual quantiles, $\Delta_{ij}(z) = q_{1-\alpha}(|r_{ij}|)$, and define $\mathcal{M}(z)$ from these bounds. Preference ambiguity is calibrated from historical weight drift using $\varepsilon = q_{1-\beta}(\|w_t - \hat{w}\|_1)$. This yields a data-driven ambiguity set that feeds directly into the robust LP core.

This loop creates an auditable bridge between predictive AI outputs and optimization decisions, and it supports transparent governance of robustness-conservatism trade-offs.

**Implementation notes.** From an engineering perspective, the pipeline is lightweight: scenario generation and coefficient precomputation are embarrassingly parallel, and the online optimization stage solves LPs with modest dimension growth in $S$. The decision artifacts are interpretable (weights, scenario constraints, regret envelopes), which simplifies model-risk review and stakeholder communication.

**Limitations in current scope.** First, uncertainty is represented through scenario approximation; quality depends on scenario coverage and calibration. Second, product-set uncertainty does not model explicit statistical dependence between KPI shocks and preference shifts. Third, additive utility may miss interaction effects between criteria (e.g., nonlinearity between lead time and defect risk). These limitations motivate the coupled and distributionally robust extensions discussed in Appendix sections.

**Practical recommendation.** When service failure penalties are severe, robust max–min is a natural default. When management prioritizes avoiding large opportunity losses versus state-wise best feasible performance, minimax regret is often preferable. A hybrid governance policy can track both metrics and switch emphasis by season or campaign phase.

## 11 CONCLUSION

We presented a tractable robust MCDA model for marketplace logistics under joint uncertainty in criteria and preferences. Using product-set uncertainty and scenario approximation, we obtained exact LP reformulations for robust max–min utility and minimax regret. Monte Carlo evaluation supports a consistent pattern: robust max–min improves worst-case guarantees, while minimax regret reduces benchmark-relative loss extremes, especially in stress conditions. The resulting methodology is computationally practical and naturally integrates into AI-assisted DSS pipelines. Future work includes coupled uncertainty sets and distributionally robust extensions.

For reproducibility details (data preprocessing, scenario generation, LP settings, and reporting items), see Appendix A.

## ACKNOWLEDGMENTS

This research was supported by a grant for the organization of training programs for leading specialists in the field of artificial intelligence, provided by the Analytical Center under the Government of the Russian Federation (Agreement No. 70-2025-000735, May 29, 2025), IGK 000000Ts330325R2Zh0002.

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

## A  REPRODUCIBILITY AND REPORTING CHECKLIST

To facilitate transparent reviewing and future replication, we summarize the reporting choices used in this submission.

**Model specification.**   All optimization objects are fully specified: feasible set $\mathcal{X}$, additive utility form, uncertainty sets $\mathcal{M}$ and $\mathcal{W}$, scenario coefficients $c^{(s)}$, and LP epigraph formulations for both robust criteria. This makes the computational core unambiguous and solver-independent.

**Experimental protocol.**   Train/test scenario separation is explicit, with repeated runs and bootstrap confidence intervals. The protocol uses fixed scenario counts per repetition and consistent evaluation metrics across all strategies. This avoids selective reporting and allows apples-to-apples comparison.

**Scope of claims.**   We intentionally report structural empirical conclusions (dominance patterns and trade-off behavior) that are reproducible under repeated Monte Carlo design, and avoid over-claiming fine-grained numeric superiority outside the tested uncertainty regimes. Extended numeric artifacts and additional plots are supplied in appendices to keep the main text policy-compliant.

**Deployment traceability.**   The framework is designed so each decision cycle can be logged by four auditable objects: (i) uncertainty-set parameters, (ii) scenario sample seed/protocol, (iii) LP solution outputs, and (iv) post-hoc diagnostics (regret profile, EVPI, VoIU). This trace enables retrospective analysis after operational incidents and supports governance requirements in industrial DSS environments.

**Reviewer-oriented summary.**   In compact form, the paper contributes: (1) a joint-uncertainty MCDA formulation, (2) exact LP reductions for two robust objectives, (3) a stress-tested empirical comparison under repeated train/test Monte Carlo, and (4) a practical DSS loop that links prediction quality, optimization robustness, and monitoring feedback.

**Operational handoff template.**   For implementation teams, a practical handoff consists of: (i) a criterion dictionary with normalization direction and update frequency, (ii) a scenario-generation specification for base/stress modes, (iii) governance-approved ranges for $\Delta$ and $\varepsilon$, (iv) objective-selection rules (max–min, minimax regret, or dual-dashboard), and (v) a monitoring playbook linking metric thresholds to intervention actions. This handoff structure reduces the gap between model design and routine operational use.

**Minimal reproducibility bundle.**   The minimum artifact bundle includes: scenario seeds and generation code, LP model definitions for equations equation 9 and equation 11, solver logs, strategy vectors for each repetition, and evaluation scripts for worst-case utility, maximum regret, EVPI, and VoIU. Publishing this bundle (or storing it in an internal audit registry) is sufficient for full end-to-end replication of the reported protocol.

## B  FULL PROOFS

### B.1  PROOF OF LEMMA 1

Consider
$$\max_{x \in \mathcal{X}} \min_{s=1,\dots,S} (c^{(s)})^\top x.$$
For fixed $x$, define $\phi(x) = \min_s (c^{(s)})^\top x$. Introduce variable $\eta$ and impose $\eta \leq (c^{(s)})^\top x$ for all $s$. Then
$$\eta \leq \min_s (c^{(s)})^\top x = \phi(x)$$
and maximizing $\eta$ forces $\eta = \phi(x)$ at optimum. Thus
$$\max_{x \in \mathcal{X}} \phi(x) = \max_{x \in \mathcal{X}, \eta} \left\{ \eta : \eta \leq (c^{(s)})^\top x, \forall s \right\},$$
which is exactly equation 9 after sign rearrangement.

## B.2 PROOF OF LEMMA 2

Scenario minimax regret is

$$\min_{x \in \mathcal{X}} \max_{s=1,\ldots,S} \left[ V_s^\star - (c^{(s)})^\top x \right].$$

Let $\psi(x) = \max_s \left[ V_s^\star - (c^{(s)})^\top x \right]$. Introduce $\rho$ with constraints $\rho \geq V_s^\star - (c^{(s)})^\top x$ for each $s$. Then $\rho \geq \psi(x)$ and minimization forces equality at optimum, yielding

$$\min_{x \in \mathcal{X}, \rho} \left\{ \rho : V_s^\star - (c^{(s)})^\top x \leq \rho, \forall s \right\},$$

which is equation 11. Linearity follows because $V_s^\star$ are constants after preprocessing.

## C EXTENDED EXPERIMENTAL PROTOCOL

**Data and normalization.** We use anonymized marketplace logistics snapshots with six criteria: cost, lead time, reliability, defect level, volatility, and disruption risk. Each criterion is normalized to $[0, 1]$ with benefit/cost orientation handled before aggregation.

**Uncertainty generation.** For each repetition, train and test scenarios are sampled independently from the same regime-specific uncertainty configuration. The base regime uses moderate perturbation radii, while stress scales both KPI intervals and preference perturbation frequency/amplitude.

**Training and evaluation.** For each $\varepsilon \in \{0, 0.05, 0.10, 0.20\}$ and each strategy class, optimization is done on $S_{\text{train}} = 500$ scenarios. Performance is evaluated on independent $S_{\text{test}} = 5000$ scenarios. Reported uncertainty over repetitions uses percentile bootstrap with $B = 5000$.

## D ADDITIONAL RESULTS AND PLOTS

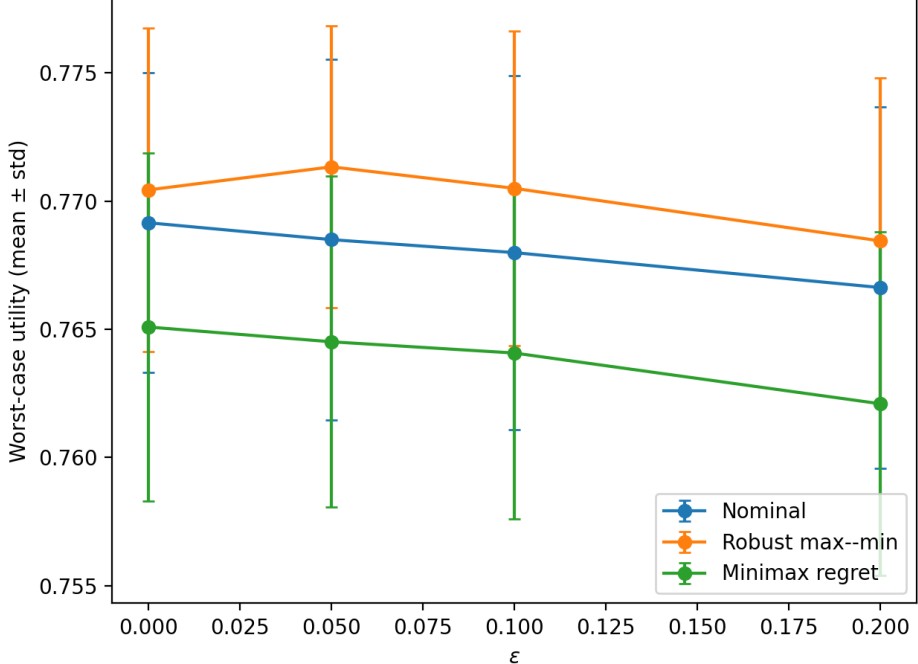

Figure 2: Out-of-sample worst-case utility in base regime.

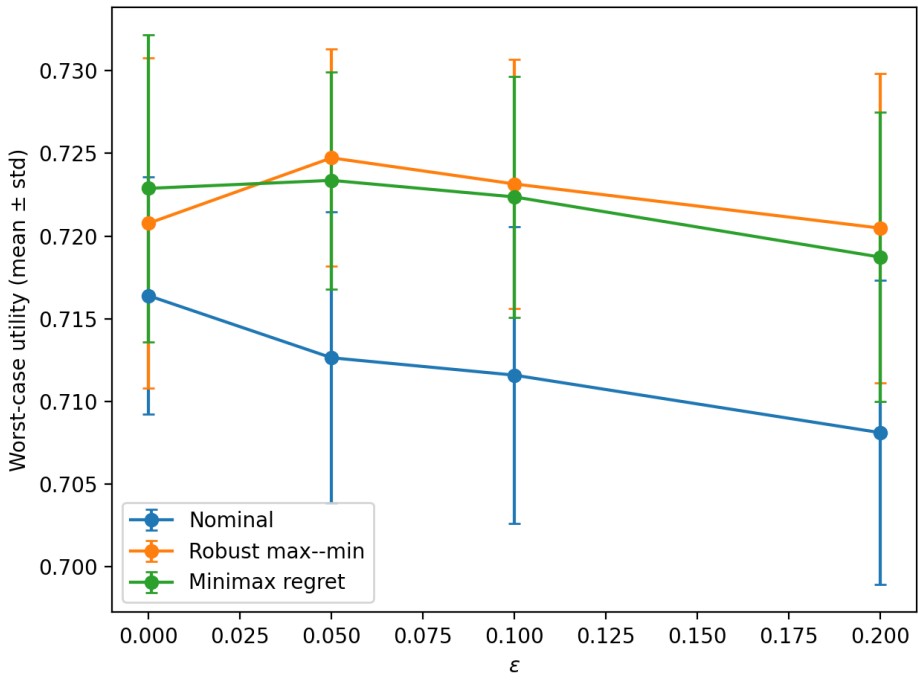

Figure 3: Out-of-sample worst-case utility in stress regime.

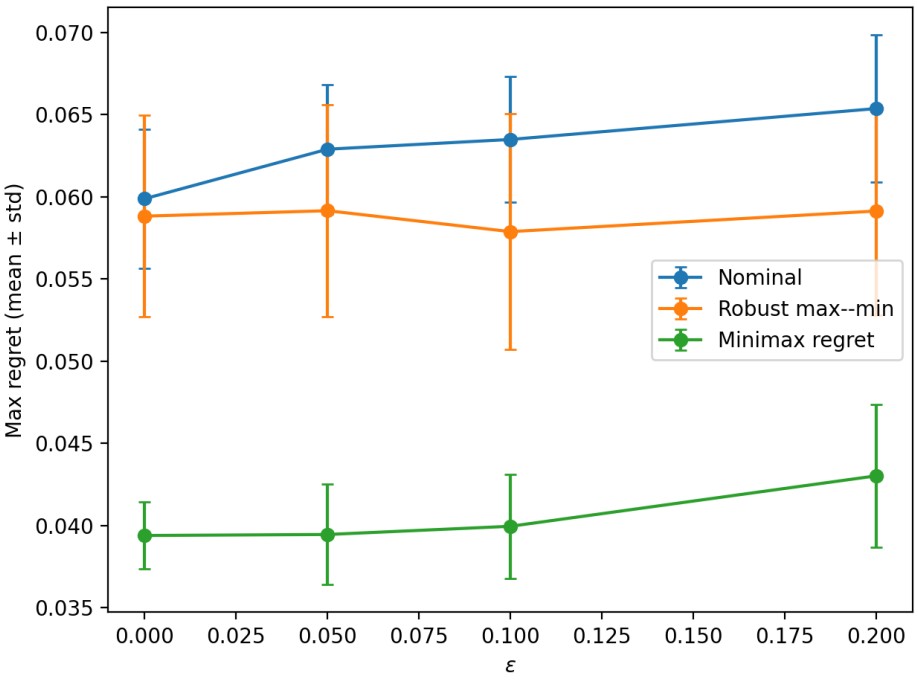

Figure 4: Out-of-sample maximum regret in base regime.

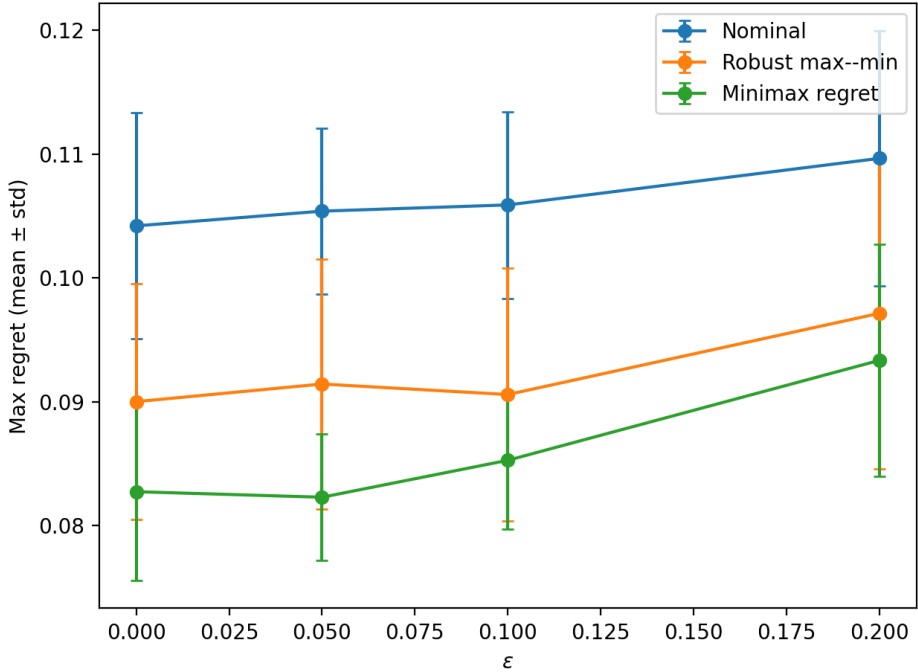

Figure 5: Out-of-sample maximum regret in stress regime.

## E    EXTENDED RELATED WORK

The full manuscript contains a broader survey across (i) deterministic MCDA ranking methods, (ii) robust preference elicitation and incomplete-weights models, (iii) robust/distributionally robust optimization in supply chains, and (iv) DSS architectures for AI-assisted decision workflows. We compressed that survey in the main text to satisfy the strict 6–10 page policy for the main body.

At a high level, deterministic MCDA methods remain useful for transparent ranking but are highly sensitive to weight perturbations and noisy indicators. Robust optimization methods provide principled worst-case guarantees via uncertainty sets (Ben-Tal & Nemirovski, 1998; Bertsimas & Sim, 2004; Kouvelis & Yu, 1997), while stochastic and DRO paradigms provide expectation/risk controls under distributional assumptions (Birge & Louveaux, 2011; Chen & Chen, 2025; Ash et al., 2022). Our contribution sits between these families: it retains MCDA interpretability while giving exact LP-solvable robust objectives under simultaneous uncertainty in $M$ and $w$.

## F    REPRODUCIBILITY NOTES

The optimization tasks are linear programs and can be solved with standard LP solvers. Scenario generation, train/test splitting, and bootstrap uncertainty estimates follow standard Monte Carlo and bootstrap practice (Rubinstein & Kroese, 2016; Efron & Tibshirani, 1993). The strict separation of train and test scenarios is maintained in every repetition to avoid optimistic bias.

