# OpenReview forum: "Robust Multi-Criteria Decision Support for AI-Assisted Marketplace Logistics with Uncertain Data and Preferences"
_mathai.club/MathAI/2026/Conference — 2026 Oral_

### Official Review · Reviewer_Df53 · 2026-03-11
**Overall Verdict: PASS**

**Rating:** 7
**Confidence:** 3

**Review:**

Key findings across all audit dimensions:
References (18/20 verified): Two minor flags — Ref 19 (Sahoo et al.) lists a phantom third author "J. Panigrahi" not found in any database, and Ref 16 (Panesh et al.) is a Russian-language article that's hard to verify but appears legitimate.
Technical Content (all correct): Every equation is well-formed, both LP reformulations are exact (verified proofs), and the ε < 2 bound on the preference ambiguity radius is mathematically elegant and necessary.
Figures & Tables (consistent): Table 1 passes all programmatic checks — monotonicity in ε, correct strategy ordering, and every numerical claim in the text matches the table exactly.

---

### Official Review · Reviewer_DXka · 2026-03-12
**The article proposes an enhanced MCDA model for logistics that integrates KPI and preference uncertainty, surpassing standard approaches with mathematical justification of variable interactions.  The research is mathematically rigorous and empirically reliable but requires simplified exposition in a financial-economic style and clarification of authorship.**

**Rating:** 6
**Confidence:** 3

**Review:**

The article proposes an enhanced multi-criteria decision-making model for logistics chains that incorporates conflicting criteria of uncertainty and KPIs in preferences.

Quality: the proposed calculations are mathematically justified, with descriptions of the variables' properties and the impact of changes on the dependent variables. The proposed multi-criteria decision-making model is supported by the conducted experiment, which revealed limitations in adaptability.

Clarity: research is conducted in a rigorous scientific writing style. It includes a mathematical description of the obtained results and calculations, which could benefit from a more detailed explanation for broader understanding.

Originality: the model proposed in the work extends beyond standard MCDM frameworks (for example TOPSIS) by offering a novel mathematical justification for variable interactions.

Significance: the results of the work fill a theoretical gap in classical deterministic MCDA models by addressing joint uncertainty through minimax utility scenario constraints. From a practical standpoint, the work proposes a step-by-step preparation of uncertainty criteria for the model and determination of acceptable input data ranges.

Pros: еhe empirical reliability of the findings, substantiated by the employed sources. The variable formulations used in the model are sufficiently well-characterized.

Cons: it would be beneficial to present the description of the model's properties and variable interactions in an additional financial-economic writing style, which could facilitate broader application in the logistics sector. In source 19, author J. Panigrahi is listed among the authors, though his works are presented in the biotechnological sectors, which are unrelated to artificial intelligence.

---

> ### Author Rebuttal · Authors · 2026-03-13
>
> We thank the reviewer for the careful reading. Regarding Reference 19 (Sahoo et al.), we confirm that the citation will be corrected to accurately reflect the authorship as it appears in the original publication. We also note the suggestion to complement the mathematical exposition with a financial-economic framing and will incorporate this perspective in the revised version to broaden accessibility for logistics practitioners

---

### Official Review · Reviewer_koVm · 2026-03-13
**Review of "A Robust Multi-Criteria Decision Model for AI-Assisted Marketplace Logistics"**

**Rating:** 7
**Confidence:** 4

**Review:**

This article presents a practical and computationally efficient framework for multi-criteria decision-making under joint uncertainty in marketplace logistics. The core innovation is a "product-set" model that separates uncertainty into interpretable components: interval-based uncertainty for KPI forecasts and an l1​-norm budget for stakeholder preferences. By using scenario approximation, the authors reformulate robust max-min utility and minimax regret into exact, tractable Linear Programs, ensuring the model is solvable at scale. The paper also provides diagnostic tools and a calibration protocol for end-to-end deployment within an AI-assisted Decision Support System.

Strengths:

1. The framework is highly practical and ready for deployment, with a clear end-to-end pipeline from predictive models to optimization.
2. The product-set uncertainty structure is interpretable and cleanly separates KPI uncertainty from preference ambiguity.
3. The exact LP reformulations ensure tractability and linear scaling with the number of scenarios.
4. Rigorous validation through Monte Carlo simulations with base and stress regimes demonstrates consistent and explainable trade-offs.

Weaknesses:

1. The model relies on the standard but often unrealistic assumption of additive utility independence between criteria.
2. The product-set formulation assumes KPI shocks and preference shifts are independent, whereas in practice they are often correlated during major disruptions.
3. Performance depends heavily on the quality of scenario generation, with limited guidance on how to generate "good" scenarios.
4. The minimax regret approach requires pre-solving numerous LPs, which can become a computational bottleneck for very large scenario sets.
5. Empirical validation lacks comparison against other robust MCDA methods or alternative uncertainty structures.

Conclusion: This is a valuable and actionable paper that makes a strong contribution to applied operations research by providing a practical, interpretable, and computationally efficient framework for robust multi-criteria decision-making in AI-assisted logistics systems.

---

### Decision · Program_Chairs · 2026-03-14

**Decision:**

Accept (Oral)

**Comment:**

Dear Author(s),

On behalf of the Program Committee of the International Conference on Mathematics of Artificial Intelligence (MathAI 2026), we are pleased to inform you that your paper has been accepted for an oral presentation at MathAI 2026.

Your paper was evaluated through a rigorous two-stage review process involving both automated screening and expert review by members of the Program Committee. The reviewers recognized the quality and contribution of your work.

Presentation details:

- Format: Oral presentation (15–20 minutes + 5 minutes Q&A)
- Mode: You may present either in person (offline) at the conference venue in Sirius, Russia, or remotely via Zoom. Please indicate your preferred mode when confirming your participation.
- Conference dates: Marh 30 - April 3, 2026
- Website: https://mathai.club

Next steps:

1. Please confirm your participation and presentation mode by replying to this email mathai.club@yandex.ru no later than March 15, 2026 18:00 Moscow time.
2. If you plan to attend in person, the organizing committee will provide accommodation details separately.
3. Please prepare your final camera-ready manuscript according to the formatting guidelines available at https://mathai.club and upload it to OpenReview by March 15, 2026 18:00 Moscow time.

Should you have any questions regarding the program, logistics, or your presentation slot, please do not hesitate to contact us.

We look forward to your contribution to MathAI 2026.

With kind regards,

MathAI 2026 Program Committee
International Conference on Mathematics of Artificial Intelligence
https://mathai.club
OpenReview: https://openreview.net/group?id=mathai.club/MathAI/2026/Conference
Telegram: https://t.me/MathAI_club
Email: mathai.club@yandex.ru